# Effects of *Piper nigrum* L. Fruit Essential Oil Toxicity against Stable Fly (Diptera: Muscidae)

**DOI:** 10.3390/plants12051043

**Published:** 2023-02-24

**Authors:** Thekhawet Weluwanarak, Tanasak Changbunjong, Arpron Leesombun, Sookruetai Boonmasawai, Sivapong Sungpradit

**Affiliations:** 1The Monitoring and Surveillance Center for Zoonotic Diseases in Wildlife and Exotic Animals (MoZWE), Faculty of Veterinary Science, Mahidol University, Nakhon Pathom 73170, Thailand; 2Department of Pre-Clinic and Applied Animal Science, Faculty of Veterinary Science, Mahidol University, Nakhon Pathom 73170, Thailand

**Keywords:** *Piper nigrum*, contact toxicity, fumigant toxicity, insecticide, stable flies, fly vector

## Abstract

The efficacy of *Piper nigrum* L. fruit essential oil (EO) against *Stomoxys calcitrans* (stable fly), a blood-feeding fly distributed worldwide, was investigated. This study aimed to evaluate the insecticidal activity of EO based on contact and fumigant toxicity tests. Chemical analysis of the EO using gas chromatography–mass spectrometry revealed that sabinene (24.41%), limonene (23.80%), β-caryophyllene (18.52%), and α-pinene (10.59%) were the major components. The results demonstrated that fly mortality increased with increasing EO concentration and time during the first 24 h of exposure. The median lethal dose was 78.37 µg/fly for contact toxicity, while the 90% lethal dose was 556.28 µg/fly. The median lethal concentration during fumigant toxicity testing was 13.72 mg/L air, and the 90% lethal concentration was 45.63 mg/L air. Our findings suggested that essential oil extracted from *P. nigrum* fruit could be a potential natural insecticidal agent for control of stable fly. To examine the insecticidal properties of *P. nigrum* fruit EO, further field trials and investigation into the efficacy of nano-formulations are required.

## 1. Introduction

Stable fly [*Stomoxys calcitrans* (Diptera: Muscidae)], an insect belonging to the order Diptera, is distributed worldwide. Stable flies carry bovine life-threatening viral pathogens including lumpy skin disease virus, bovine herpes virus, and bovine leukosis virus [1]. Furthermore, in the equine industry, stable fly acts as a mechanical vector of equine infectious anemia virus [1,2] and *Trypanosoma evansi* [3]. Moreover, stable fly plays an important role as the biological vector of the equine stomach nematode *Habronema microstoma* [4]. This vector has been demonstrated to cause economic impact in equine and livestock husbandry by disease transmission, annoyance, and stress in affected animals [1,5].

Various chemical products have been used to control insect vectors. In an experiment involving stable flies, the toxicity of semiochemicals including aliphatic alcohols, terpenoids, ketones, and carboxylic esters was studied, with the median lethal concentration (LC_50_) ranging from 16.30–40.41 μg/cm^2^ [6]. Recently, cypermethrin (group A) demonstrated an excellent repellent effect with an LC_50_ of 1.52 μg/mL [7]. However, the emergence of flies resistant to these chemical substances has been reported worldwide [8,9], and furthermore the chemical residue causes environmental pollution [10].

To control diseases caused by the stable fly vector using green or environmentally friendly products, plant essential oil (EO) has been proposed as an alternative method for insecticidal and repellent insect-vector control. By its volatile properties, EO affects the flight behavior of stable fly and peripheral olfactory networks in the antennae [11]. A repellency study in stable fly using a human hand bioassay demonstrated that various EOs such as patchouli [*Pogostemon cablin* (Blanco) Bentham], clove (*Eugenia caryophyllata* Thunberg) buds and leaves, and lovage (*Levisticum officinale* L. Koch) roots, had the potential to repel female flies [12]. Catnip EO (*Nepeta cataria* L.), whose main active ingredient is nepetalactone, demonstrated feeding and oviposition repellence against stable flies [13]. Recently, *Melaleuca alternifolia* EO, with a high 1,8-cineole component, was investigated for its insecticidal activity against adult stable flies, using surface application and oil-impregnated paper exposure testing [14].

The insecticidal effects of potential medicinal plant essential oils have been tested against stable fly. Catnip EO (90% ZE- and EZ-nepetalactone and 10% caryophyllene) had an LC_50_ of 7.7 mg/cm^3^ and an LC_90_ of 18.15 mg/cm^3^ for contact toxicity, and an LC_50_ of 10.7 mg/cm^3^ and an LC_90_ of 23.90 mg/cm^3^ for fumigant toxicity [15]. Recently, bitter orange [*Citrus aurantium* (L.)] EO, which primarily contains limonene, had a reported LD_50_ of 105.88 µg/fly and an LD_90_ of 499.25 µg/fly for contact toxicity, and an LC_50_ of 13.06 mg/L air and an LC_90_ of 43.13 mg/L air for fumigant toxicity within 24 h [16]. Additionally, the semiochemicals rosalva, geranyl acetone, and citronellol were studied for their fumigant toxicity in stable fly, using a residual contact bioassay in glass jars. The LC_50_ of rosalva (from the rose), geranyl acetone (from cardamom, citrus, and petitgrain oil), and citronellol (from eucalyptus) were 13.10 µg/cm^2^, 25.20 µg/cm^2^, and 35.69 µg/cm^2^, respectively, while the LC_90_ values were 18.54 µg/cm^2^, 34.97 µg/cm^2^, and 50.10 µg/cm^2^, respectively, after 24 h exposure [6]. Overall, however, the study of potential volatile plant oil for controlling the stable fly insect vector has been limited.

*Piper nigrum* L. is widely employed as a food ingredient to increase taste and is found mainly in tropical zones throughout the world [17]. A previous study reported that green or unripe pepper EO had stronger antibacterial and antifungal properties compared with black pepper EO [18]. The EO from green pepper was analyzed and found to have the same primary components including β-pinene, δ-3-carene, limonene, α-pinene, and caryophyllene oxide [18]. Another study found that air-dried green pepper EO consisted primarily of terpinen-4-ol, β-caryophyllene, hedycaryol, limonene, and sabinene. Moreover, green pepper oil contained the highest levels of sesquiterpenes such as β-caryophyllene and oxygenated terpenoids such as caryophyllene oxide, hedycaryol, β-eudesmol, and eugenol when compared with black and white peppers [19].

Black pepper oil contained primarily monoterpene hydrocarbons (47–64%) and sesquiterpene hydrocarbons (30–47%) [20]. The EO from black pepper was analyzed, and the primary components were found to be β-caryophyllene, limonene, β-pinene, α-pinene, δ-3-carene, sabinene, and myrcene [21]. A study of black pepper components found that piperine, an alkaloid, had multiple pharmacological effects, including insecticidal, larvicidal, antibacterial, antifungal, and antioxidant activity [22], while β-caryophylline, a sesquiterpene, demonstrated a larvicidal effect on *Aedes aegypti* mosquito larvae [23]. Fumigant toxicity and the biochemical effects of *P. nigrum* EO were studied in the adult and larval stages of the pulse beetle *Callosobruchus chinensis* (Coleoptera: Bruchidae) [24] and the adult red flour beetle *Tribolium castaneum* (Coleoptera: Tenebrionidae) [25]. However, study of green pepper fruit EO, which has a different chemical composition, remains limited.

The aim of the present study was to evaluate the effects of unripe or green *P. nigrum* fruit EO insecticidal activities against *S. calcitrans*, using in vitro contact toxicity and fumigant toxicity methods. Thereafter, the EO was analyzed for its chemical composition using gas chromatography–mass spectrometry (GC-MS).

## 2. Results

### 2.1. Essential Oil Extraction and Quantification

The yield of *P. nigrum* L. EO obtained from fresh fruit was 0.96% volume/weight (*v/w*). The oil was clear in color with a pH of 5, a density of 0.87 g/mL at 20 °C, and a refractive index of 1.48. The chemical constituents of *P. nigrum* EO were determined by GC-MS techniques. A total of twenty-one compounds were identified accounting for 99.41% of the total oil including the retention time (min), classes, compounds, formula, chemical structure, peak area (%), and % similarity index (Table 1). A chromatogram of the primary components of *P. nigrum* EO is shown in Figure 1. The major components were sabinene (24.41%), limonene (23.80%), β-caryophyllene (18.52%), and α-pinene (10.59%). Limonene and β-caryophyllene were used to quantify the EO, and the concentrations were 0.18% and 0.10% weight/volume (*w/v*), respectively.

### 2.2. Contact Toxicity Test

The contact toxicity of *P. nigrum* EO against *S. calcitrans* was studied at various EO concentrations. The test was validated using acetone, with no insecticidal activity, as a negative control. When EO treatments at 43.25 µg/µL, 86.50 µg/µL, 216.25 µg/µL, and 432.50 µg/µL were compared with cypermethrin, EO was found to have little to no insecticidal efficacy (Table 2). From 1–24 h after treatment, EO at a concentration of 865.00 µg/µL demonstrated insecticidal activity comparable to cypermethrin. The results showed that *S. calcitrans* mortality increased with higher EO concentration and time. The interaction between EO concentration and time was statistically significant for *S. calcitrans* mortality (time, F_(2.35, 33.01)_ = 32.61, *p* < 0.001; treatment, F_(6, 33.01)_ = 244.77, *p* < 0.001; treatment × time, F_(14.15, 33.01)_ = 4.64, *p* < 0.001). The LD_50_ and LD_90_ values were 15.68% and 111.23% (*w*/*v*) or 78.37 µg/fly and 556.28 µg/fly, respectively (Table 3).

### 2.3. Fumigant Toxicity Test

The fumigant toxicity of *P. nigrum* EO against *S. calcitrans* was assessed at various EO concentrations. The test was validated using acetone, which has no insecticidal activity, as a negative control. At EO concentrations < 25.95 mg/L, the EO had little or no insecticidal action for *S. calcitrans* (Table 4). From 6–24 h after treatment, the EO at a concentration of 25.95 mg/L demonstrated insecticidal activity comparable to cypermethrin. The results showed that *S. calcitrans* mortality increased with the increased EO concentration and time. The interaction between concentration and time was statistically significant for *S. calcitrans* mortality (time, F_(2.28, 31.97)_ = 75.47, *p* < 0.001; treatment, F_(6, 31.97)_ = 158.11, *p* < 0.001; treatment × time, F_(13.70, 31.97)_ = 24.20, *p* < 0.001). The LC_50_ and LC_90_ values were 13.72 mg/L air and 45.63 mg/L air, respectively (Table 5).

## 3. Discussion

This investigation utilized *P. nigrum* fruit EO for in vitro contact and fumigant toxicity testing against wild-caught stable flies from a rural horse farm. Since the wild-caught flies varied in age, sex, and nutritional and physiological status, these factors should be considered when comparing the results from this study with those of laboratory studies [26]. While the laboratory-reared stable fly is suitable for insecticide-sensitivity and vector-competency experiments under controlled conditions [1,27], the field-caught stable fly might better represent the actual response to the plant’s insecticidal toxicity under real field conditions [16,28].

Previous reports have described the repellent and insecticidal effects of *P. nigrum* EO. The fumigation activity of *P. nigrum* EO has been reported against *Tribolium castaneum* (Coleoptera: Tenebrionidae) [25,29]. EOs obtained from black pepper had repellent, insecticidal, oviposition inhibitory, and acetylcholinesterase (AChE) enzyme inhibitory activities against maize weevil, *Sitophilus zeamais* [30]. A fumigation toxicity assay measured an LC_50_ of 0.287 μL/cm^3^ and 0.152 μL/cm^3^ after 24 and 48 h exposure periods, and a contact toxicity assay reported an LC_50_ of 0.208 μL/cm^2^ and 0.126 μL/cm^2^ after 24 and 48 h exposure periods, respectively [30]. In addition, chitosan-encapsulated nanoparticles of *P. nigrum* EO demonstrated insecticidal activity via the AChE function against insect pest species *T. castaneum* and *Sitophilus oryzae* [31].

In this study, unripe *P. nigrum* fruit EO was shown to have insecticidal effects, via contact and fumigant toxicity studies. The four most important chemical compounds identified in this study were sabinene (24.41%), limonene (23.80%), β-caryophyllene (18.52%), and α-pinene (10.59%), which altogether represented approximately 77.32% of all *P. nigrum* EO. Sabinene, limonene, and α-pinene belong to the monoterpene group, which is the predominant class of terpene found in *P. nigrum* oil [32], while β-caryophyllene belongs to the sesquiterpene group [22]. The chemical composition of *P. nigrum* fruit EO in this study was similar to that in other reports [21]. However, differences in chemical composition are likely; EO obtained from black pepper grown in Madagascar and Brazil had varying compositions of sabinene (0.1–13.9%), limonene (15.1–38.1%), β-caryophyllene (<0.05–14.8%), and α-pinene (5.1–28.7%) [21]. Another study of *P. nigrum* fruit cultivated in Brazil found the main volatile compounds in the monoterpene group to be α-pinene (6.91–10.67%), β-pinene (22.61–35.05%), and limonene (21.0–31.77%) [33]. Thus, various geographic locations, seasons of the year, and extraction methods all affect the chemical composition of the EO [21].

Medicinal plant EOs containing at least two of the major compounds sabinene, limonene, β-caryophyllene, or α-pinene have been reported to have insecticidal effects. *Xylopia aethiopica* EO containing sabinene (26.1 ± 3.1%), β-pinene (17.4 ± 1.9%), and α-pinene (9.6 ± 1.6%) had contact toxicity against the house fly (*Musca domestica*) and exhibited an LD_50_ of 61.5 µg/adult and LC_90_ of 178.5 µg/adult in males, and an LD_50_ of 30.7 µg/adult and LC_90_ 164.2 µg/adult in females [34]. When sabinene was applied as a single component to the female *M. domestica*, there was a higher LD_50_ of 109.7 µg/adult and LC_90_ of 213.8 µg/adult. This was similar to the results when α-pinene was applied to the female *M. domestica*, with a higher LD_50_ of 69.7 µg/adult and LC_90_ of 254.7 µg/adult. In contrast to its effects in females, the LD_50_ and LC_90_ values of sabinene decreased to 10.4 µg/adult and 117.5 µg/adult in male *M. domestica*, and the LD_50_ and LC_90_ of α-pinene decreased to 8.6 µg/adult and 100.2 µg/adult, respectively. This suggests that major and minor compounds, acting synergistically or antagonistically, are responsible for the insecticidal activity [34]. EO extracted from *Schinus areira* seeds containing the monoterpenes sabinene (39.15%), α-pinene (14.96%), and β-pinene (8.75%) also showed insecticidal activity against the seed beetle, *Rhipibruchus picturatus*. Using a molecular docking assay, the main component from *S. areira* leaves was found to bind the AChE enzyme’s active site [35]. *Aloysia citriodora* EO containing sabinene (22.9%), limonene (7.4%), and caryophyllene (2.4%) was evaluated against the southern green stinkbug *Nezara viridula* (L.), using fumigant activity in an enclosed chamber. The toxicity of *A. citriodora* increased with increasing concentration and exposure time. Fumigant activity had an LC_50_ of 13.5 μg/mL air 24 h after treatment and contact toxicity had an LC_50_ of 8.1 μg/cm^2^ [36]. *Haplopappus foliosus* EO consisted of limonene (28.0%), caryophyllene (3.97%), β-pinene (1.43%), sabinene (0.93%), and α-pinene (0.89%), with an LC_50_ value of 4.43 mg/dm^3^ air against *M. domestica* [37].

This study noted a contact toxicity LD_50_ of 78.37 µg/fly and an LD_90_ of 566.28 µg/fly 24 h after exposure, which differs slightly from the toxicity of citrus EO for the stable fly (LD_50_ of 105.88 µg/fly and LD_90_ of 499.25 µg/fly). However, the fumigant toxicity values of an LC_50_ of 13.72 mg/L air and an LD_90_ of 45.63 mg/L air were consistent with a previous report (LC_50_ of 13.06 mg/L air and LC_90_ of 43.13 mg/L air) [16]. Furthermore, the percentage of mortality from citrus EO at 25.20 mg/L air was similar to that of 1% cypermethrin, from 2–24 h after treatment. This was somewhat better than the results of the current study (25.95 mg/L air from 6–24 h after treatment). Limonene (93.79%) is the major compound found in citrus EO [16], while sabinene (24.41%), limonene (23.80%), β-caryophyllene (18.52%), and α-pinene (10.59%) are the major compounds found in *P. nigrum* EO. Prior studies have reported the insecticidal effects of compounds found in essential oil. Sabinene is reportedly insecticidal against *Aphis pomi* [38], has a repellent effect against *Tribolium castaneum* [39], and exhibits strong fumigant toxicity against maize weevils [40]. Limonene has insecticidal activity against mealybugs and scale insects [41], horn flies [42], and the tomato leaf miner [43]. Combinations of these compounds can exert insecticidal activity. Sabinene has additive effects with α-pinene, 1,8-cineole, 1-octen-3-ol, and linalool, and limonene has a synergistic effect with α-pinene and sabinene [44]. β-caryophyllene and α-pinene are toxic to *Aphis gossypii* Glover (Homoptera: Aphididae) by inhibiting AChE activity, polyphenol oxidase, carboxylesterase, and glutathione S-transferases in *A. gossypii* [45].

The cytotoxicity of *P. nigrum* EO has been investigated on a normal cell line, human fetal lung fibroblast (MRC-5), and five different tumor cell lines including human cervical carcinoma (HeLa), human myelogenous leukemia (K562), lung adenocarcinoma (A549), colon cancer (LS-174), and melanoma (FemX). The results indicate that *P. nigrum* EO exhibited lower cytotoxic activity, with an IC_50_ of 25.57–56.74 µg/mL after 48 h of treatment compared with cisplatin (IC_50_, 2.34–9.08 µg/mL) as a positive control [46]. The toxicity of *P. nigrum* EO at concentrations of 5, 10, or 50 mg/kg was evaluated for 21 days in adult male albino Swiss mice. None of the *P. nigrum* EO doses induced signs of toxicity, such as irritability, reaction to contact, gripping the tail, twisting, writhing, grip strength, ataxia, tremor, convulsion, lacrimation, piloerection, changes in respiratory frequency, changes in body weight, or abnormalities in biochemical parameters [47]. Based on the LD_90_ of 556.28 (325.89–1475.26) µg/fly and the LC_90_ of 45.63 mg/L air obtained in the current study, we expected that the *P. nigrum* EO used in this study should not affect horses and livestock, the primary hosts of the stable fly.

Although this study did not investigate the molecular mechanism of EO effect, the insecticidal activities of EO have been reported. Plant-derived EOs, primarily monoterpenes, exhibit their neurotoxic effects by the inhibition of AChE in the octopamine pathway in insects, alongside the essential detoxification enzymes glutathione S-transferase and catalase. They are also capable of inhibiting gamma-aminobutyric acid receptors [25,48,49]. However, *P. nigrum* EO and the compounds sabinene, limonene, β-caryophyllene, and α-pinene should be the subjects of future studies of the EO insecticidal mechanism of action.

## 4. Conclusions

This study demonstrated the toxicity of unripe *P. nigrum* fruit EO, primarily containing sabinene, limonene, β-caryophyllene, and α-pinene, as an environmentally friendly insecticidal agent against stable fly (*S. calcitrans*), using contact and fumigant toxicity tests. The synergistic effects of the components of this EO on the insect’s neuromuscular system should be investigated further. As there are limited reports available discssing the insecticidal activities of *P. nigrum* fruit EO on horse and livestock vectors such as horse fly, horn fly, and midges, the study of *P. nigrum* fruit EO should be expanded and its nano-formulation for open field trials should be implemented.

## 5. Materials and Methods

### 5.1. Ethical Statement

This study protocol was approved by the Faculty of Veterinary Science, Mahidol University Animal Care and Use Committee (Ref. MUVS-2020-12-63).

### 5.2. Insects

Populations of *S. calcitrans* used in this study were collected using Nzi Traps [50], from a horse farm in Nakhon Pathom Province (13°45′43.4″ N, 100°08′15.7″ E), between March and May 2021. To the best of our knowledge, insecticides had never been used on that farm. The traps were set up at the collection site from 16:00 to 19:00. All flies were stored in plastic cups and transported in Styrofoam boxes containing ice packs to the Pharmacology Laboratory, Faculty of Veterinary Science, Mahidol University. Male and female *S. calcitrans* were tested and were selected from groups of undamaged flies under a stereomicroscope (SMZ745, Nikon, Tokyo, Japan). Blood-fed flies were separated out of this study. The samples were preserved between 27–29 °C and 70–80% relative humidity upon arrival at the laboratory until testing.

### 5.3. Essential Oil Extraction and Quantification

*Piper nigrum* L. fruit (Figure 2) was obtained from Nakhon Pathom Province in central Thailand (13°45′12.3″ N, 100°16′13.2″ E) and was pesticide-free. The plant was identified and delivered to the Department of Pharmaceutical Botany, Faculty of Pharmacy, Mahidol University (PBM No.005504-6). The EO was extracted from 50 kg of fruit, within 6 h of arrival, by the steam distillation process, where the fruits were immersed in water and the system was heated to 100 °C. The EO produced was stored in amber bottles at 4 °C until use. The EO yield is expressed in % (*v/w*) based on the fresh plant material weight.

The EO’s physical properties were analyzed as follows: color was visually verified by three different individuals, pH was measured with pH-indicator strips (Merck, Darmstadt, Germany), density was measured with a density meter (DA-100M, Tokyo, Japan), and the refractive index was calculated with a refractometer (RX-5000CX, Atago, Tokyo, Japan).

The EO’s chemical composition was determined using a GC-MS model 7890A-MS5975C (Agilent Technologies, Santa Clara, CA, USA) equipped with a DB-5HT capillary column (length, 30 m; inner diameter, 0.25 mm; film thickness, 0.1 µm). The sample was injected in split mode with a 1:10 split ratio. Helium was used as the carrier gas, with a flow rate of 1 mL·min^−1^. The injection port temperature was 250 °C, and the column temperature schedule was as follows: 50 °C for 2 min followed by a rise to 250 °C at a rate of 10 °C per minute, then maintained at 250 °C for 5 min. The MS included an ion source temperature of 230 °C, an ionization energy of 70 eV, and a mass scan range of 350–550 amu. Components were identified by comparing their mass spectra with data from the Wiley 7th edition mass spectrometry library. Concentrations of the principal components were calculated by comparing the peak area of the sample with the peak area of the standard.

### 5.4. Contact Toxicity Test

Contact toxicity of the EO from *P. nigrum* L. to *S. calcitrans* was achieved by topical application as described by Leesombun et al., with some modifications [22]. Preliminary studies were performed to determine appropriate test range concentrations. The various concentrations of *P. nigrum* EO were prepared in a 1.5 mL microcentrifuge tube using acetone as a solvent. Five EO concentrations were used in this trial: 43.25 µg/µL, 86.50 µg/µL, 216.25 µg/µL, 432.50 µg/µL, and 865.00 µg/µL, representing 5%, 10%, 25%, 50%, and 100% (*v*/*v*) of *S. calcitrans* (mixed sexes). Flies were anesthetized at −20 °C for 30–45 s, and 0.5 µL of EO at each concentration was applied directly to the fly’s chest using a micropipette. Acetone and cypermethrin (1% *w*/*v*) were applied at the same volume for negative and positive controls, respectively. Each treatment was performed in three iterations on 10 flies each time. After application, the treated flies were placed in a clear plastic cup (10 flies/cup), and the cup was covered with a fine mesh fabric and fastened with rubber bands. A honey solution (10%) on cotton was layered over the mesh fabric, for the flies to consume. The flies were brought to recovery at 27–29 °C and 70–80% relative humidity. Mortality was recorded at 1, 2, 4, 6, 12, and 24 h after treatment. Flies were determined to be dead when they did not move after being mechanically stimulated with a brush.

### 5.5. Fumigant Toxicity Test

Fumigant toxicity of the EO from *P. nigrum* active against *S. calcitrans* was examined according to the procedures reported by Leesombun et al., with some modifications [22]. This test was performed in a 1 L clear plastic container with a lid. Preliminary studies were performed to determine the appropriate range of test concentrations. Different amounts of EO (0.47, 0.93, 1.87, 2.8, and 4.67 mg) dissolved in 100 µL of acetone were pipetted separately onto a 55-mm diameter Whatman No. 1 filter paper (GE Healthcare, Buckinghamshire, UK) and placed on the bottom of a glass petri dish (55 mm diameter). The solvent on each filter paper in the petri dish was allowed to evaporate for 2–3 min, and then the petri dish was covered with a fine mesh fabric which was fastened with rubber bands to prevent contact between the filter paper and the flies. Acetone and cypermethrin (1% *w/v*) were used as negative and positive controls, respectively. The petri dishes were placed on the bottom of the plastic container. Honey solution (10%) on cotton wool was placed on the bottom of the plastic container. Mixed sexes of *S. calcitrans* were used for testing. Flies were anesthetized at −20 °C for 30–45 s, placed in the plastic containers, and each container was closed. Each treatment was performed with ten flies in 3 iterations. Flies were allowed to recover and were maintained at 27–29 °C with 70–80% relative humidity. Mortality was recorded at 1, 2, 4, 6, 12, and 24 h after treatment. Flies were determined to be dead when they did not move.

### 5.6. Data Analysis

Insect mortality was calculated and the mortality data were corrected using Abbott′s formula [51], when the control mortality ranged between 5–20%. Normality and homogeneity of the variables were verified using the Shapiro–Wilk and Levene tests, respectively. Statistical comparisons of mortality between treatments were analyzed by one-way analysis of variance (ANOVA) followed by post hoc analysis (Tukey’s honest significant difference test), in SPSS software version 21.0 (SPSS, Chicago, IL, USA). Repeated ANOVA and Greenhouse–Geisser corrections were applied to compare the effects of the treatment concentrations and exposure times on mortality, using SPSS version 21.0 software. Exposure interval was the repeated factor and insect mortality was the response variable, with treatment as the main effect. A *p* value < 0.05 was considered significant. Probit analysis to calculate toxicity values, mean lethal dose (LD_50_) and 90% lethal dose (LD_90_) at 24 h after treatment, and mean lethal concentration (LC_50_) and 90% lethal concentration (LC_90_) at 24 h after processing was performed using LdP series software (Ehab Mostafa Bakr, Dokki, Cairo, Egypt). The software can be downloaded free of charge from http://www.ehabsoft.com/ldpline/ and was accessed on 6 November 2022.

## Figures and Tables

**Figure 1 plants-12-01043-f001:**
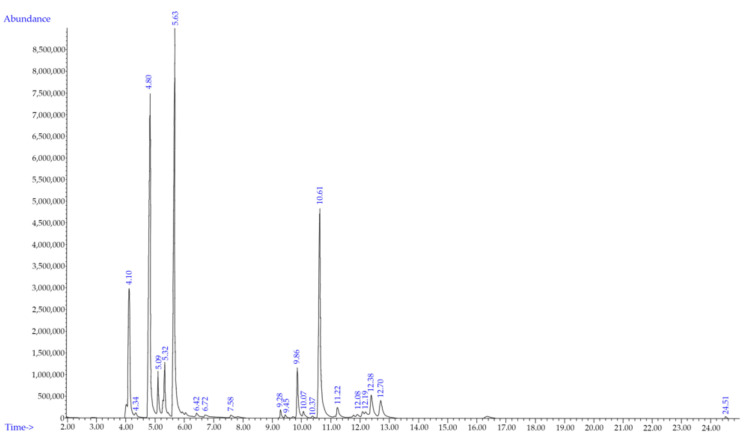
Gas chromatography–mass spectrometry chromatogram of *Piper nigrum* L. essential oil. The major peaks were α-pinene, sabinene, limonene, and β-caryophyllene, respectively.

**Figure 2 plants-12-01043-f002:**
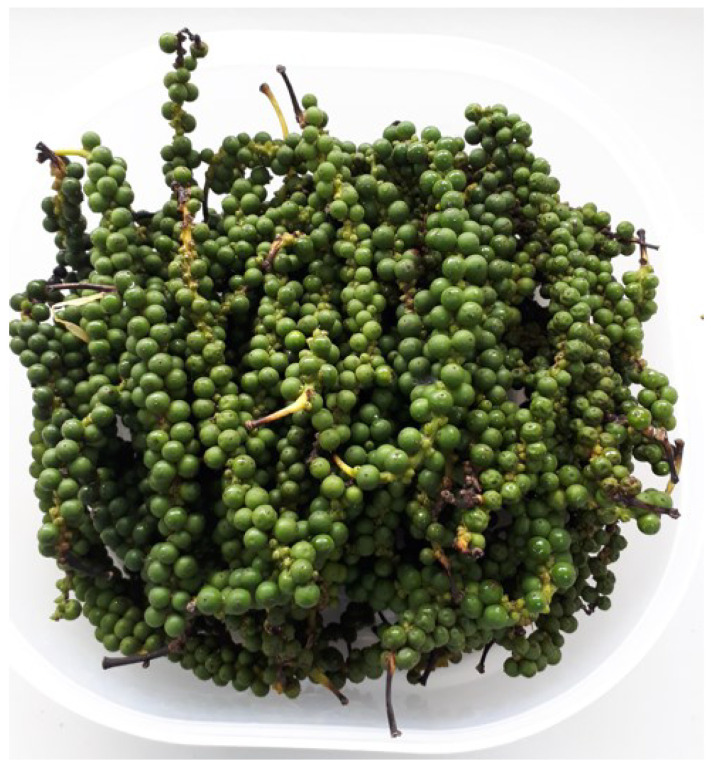
*Piper nigrum* L. fruit used in this study.

**Table 1 plants-12-01043-t001:** Chemical composition of *Piper nigrum* L. essential oil.

No.	Retention Time (min)	Classes	Compounds	Formula	Chemical Structure	Peak Area (%)	% Similarity Index
1	4.10	Monoterpene	α-Pinene	C_10_H_16_	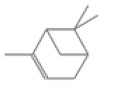	10.59	96
2	4.34	Monoterpene	Camphene	C_10_H_16_	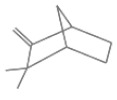	0.53	91
3	4.80	Monoterpene	Sabinene	C_10_H_16_	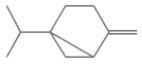	24.41	94
4	5.09	Monoterpene	β-Myrcene	C_10_H_16_	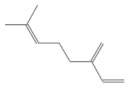	2.69	97
5	5.32	Monoterpene	3-Carene	C_10_H_16_	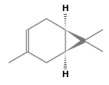	4.20	97
6	5.63	Monoterpene	Limonene	C_10_H_16_	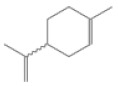	23.80	97
7	6.42	Monoterpene	α-Terpinolene	C_10_H_16_	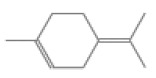	0.39	97
8	6.72	Monoterpene	L-Linalool	C_10_H_18_O	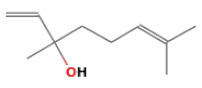	0.34	80
9	7.58	Monoterpene	4-Terpineol	C_10_H_18_O	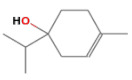	0.30	83
10	9.28	Sesquiterpene	δ-Elemene	C_15_H_24_	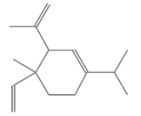	0.56	99
11	9.45	Sesquiterpene	α-Cubebene	C_15_H_24_	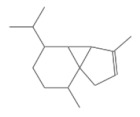	0.24	98
12	9.86	Sesquiterpene	Copaene	C_15_H_24_	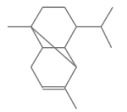	3.48	99
13	10.07	Sesquiterpene	Germacrene D	C_15_H_24_	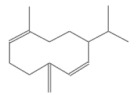	0.66	96
14	10.37	Sesquiterpene	α-Gurjunene	C_15_H_24_	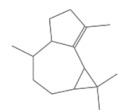	0.14	93
15	10.61	Sesquiterpene	β-Caryophyllene	C_15_H_24_	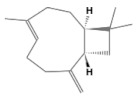	18.52	99
16	11.22	Sesquiterpene	β-Selinene	C_15_H_24_	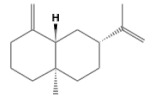	1.34	97
17	12.08	Sesquiterpene	γ-Elemene	C_15_H_24_	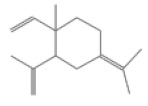	0.62	81
18	12.19	Sesquiterpene	α-Muurolene	C_15_H_24_	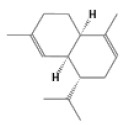	0.61	95
19	12.38	Sesquiterpene	β-Bisabolene	C_15_H_24_	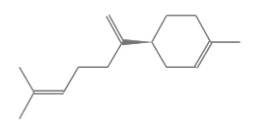	3.06	98
20	12.70	Sesquiterpene	δ-Cadinene	C_15_H_24_	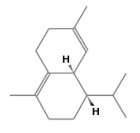	2.75	98
21	24.51	Phthalates	Bis(2-ethylhexyl) phthalate	C_24_H_38_O_4_	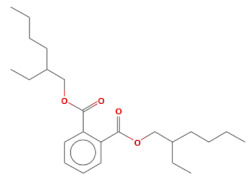	0.18	91
Total						99.41	

**Table 2 plants-12-01043-t002:** Insecticidal activity of *Piper nigrum* L. essential oil against *Stomoxys calcitrans* by contact toxicity testing.

Concentration	Mortality (%, ± SD)					
	1 h	2 h	4 h	6 h	12 h	24 h
Negative control	0 ^a^	0 ^a^	0 ^a^	0 ^a^	0 ^a^	0 ^a^
Cypermethrin (1%)	100 ^d^	100 ^d^	100 ^d^	100 ^d^	100 ^d^	100 ^e^
*P. nigrum* EO (µg/µL)						
43.25	0 ^a^	0 ^a^	6.7 ± 5.8 ^a^	6.7 ± 5.8 ^a^	13.3± 5.8 ^ab^	23.3± 5.8 ^b^
86.50	20.0 ± 0.0 ^b^	26.7 ± 5.8 ^b^	26.7 ± 5.8 ^b^	30.0 ± 10.0 ^b^	30.0 ± 10.0 ^b^	36.7 ± 11.6 ^bc^
216.25	36.7 ± 5.8 ^c^	40.0 ± 0.0 ^c^	40.0 ± 0.0 ^bc^	40.0 ± 0.0 ^bc^	50.0 ± 0.0 ^c^	53.3 ± 5.8 ^cd^
432.50	36.7 ± 5.8 ^c^	46.7 ± 5.8 ^c^	50.0 ± 10.0 ^c^	56.67 ± 11.6 ^c^	60.0 ± 10.0 ^c^	63.3± 5.8 ^d^
865.00	99.3 ± 5.8 ^d^	93.3 ± 5.8 ^d^	96.7 ± 5.8 ^d^	96.7 ± 5.8 ^d^	96.7 ± 5.8 ^d^	96.7 ± 5.8 ^e^
df	6, 14	6, 14	6, 14	6, 14	6, 14	6, 14
F	352.111	341.778	167.500	111.333	118.417	107.333
*p*	<0.001	<0.001	<0.001	<0.001	<0.001	<0.001

Post hoc analysis: significant differences (*p* < 0.05) between treatments are indicated by different letters.

**Table 3 plants-12-01043-t003:** Lethal dose (LD_50_ and LD_90_) of *Piper nigrum* L. essential oil against *Stomoxys calcitrans* according to contact toxicity testing 24 h after treatment.

Treatment	Contact Toxicity Test
LD_50_ [µg/fly] (95% CL)	78.37 (57.83–110.16)
LD_90_ [µg/fly] (95% CL)	556.28 (325.89–1475.26)
Slope ± SE	1.51 ± 0.25
χ2	5.16

95% CL, 95% confidence limit; SE, standard error; χ2 = chi-square.

**Table 4 plants-12-01043-t004:** Insecticidal activity of *Piper nigrum* L. essential oil against *Stomoxys calcitrans* by fumigant toxicity testing.

Concentration	Mortality (%)					
	1 h	2 h	4 h	6 h	12 h	24 h
Negative control	0 ^a^	0 ^a^	0 ^a^	0 ^a^	0 ^a^	0 ^a^
Cypermethrin (1%)	100 ^b^	100 ^c^	100 ^c^	100 ^c^	100 ^c^	100 ^c^
*P. nigrum* EO (mg/L air) 0.87						
	0 ^a^	0 ^a^	0 ^a^	0 ^a^	3.3 ± 5.8 ^a^	3.3 ± 5.8 ^a^
4.33	0 ^a^	3.3 ± 5.8 ^a^	6.7 ± 5.8 ^a^	6.7 ± 5.8 ^a^	10.0 ± 0.0 ^a^	10.0 ± 0.0 ^a^
8.65	0 ^a^	3.3 ± 5.8 ^a^	3.3 ± 5.8 ^a^	13.3 ± 15.3 ^a^	13.3 ± 15.3 ^a^	16.7 ± 15.3 ^a^
17.30	0 ^a^	0 ^a^	6.7 ± 5.8 ^a^	33.3 ± 5.8 ^b^	36.7 ± 11.6 ^b^	50.0 ± 17.3 ^b^
25.95	3.3 ± 5.8 ^a^	20.0 ± 10.0 ^b^	66.7 ± 5.8 ^b^	90.0 ± 0.0 ^c^	93.3 ± 5.8 ^c^	93.3 ± 5.8 ^c^
df	6, 14	6, 14	6, 14	6, 14	6, 14	6, 14
F	891.000	170.733	53.842	128.037	88.308	63.741
*p*	<0.001	<0.001	<0.001	<0.001	<0.001	<0.001

Post hoc analysis significant differences (*p* < 0.05) between treatments are indicated by different letters.

**Table 5 plants-12-01043-t005:** Lethal concentration (LC_50_ and LC_90_) of *Piper nigrum* L. essential oil against *Stomoxys calcitrans* by fumigant toxicity testing 24 h after treatment.

Treatment	Fumigant Toxicity Test
LC_50_ [mg/L air] (95% CL)	13.72 (5.71–79.85)
LC_90_ [mg/L air] (95% CL)	45.63 (n/a)
Slope ± SE	2.46 ± 0.43
χ2	28.95

95% CL, 95% confident limit; SE, standard error; χ2, chi-square; n/a, not available.

## Data Availability

The data presented in this study are available within the article.

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
