# Peer review of "Effects of Piper nigrum L. Fruit Essential Oil Toxicity against Stable Fly (Diptera: Muscidae)"

_plants, 2023, doi:10.3390/plants12051043_

Round 1
Reviewer 1 Report
The study “Effects of Piper nigrum L. fruit essential oil toxicity against stable fly (Diptera: Muscidae)” by T. Weluwanarak et al. tries to evaluate the insecticidal activity against Stomoxys calcitrans of the essential oil based on contact and fumigant toxicity tests. The composition of the studied EO was established by GS-MS and revealed the presence of four major components: sabinene, limonene, β-caryophyllene and α-pinene, respectively.
The experimental results are clearly presented; moreover, the authors discuss and compare their own data with those of the literature. Therefore, the Discussion part confers an additional value to the manuscript.
Nevertheless, the manuscript contains a few writing errors:
Page 1: Line 4 – 2nd paragraph of the Introduction: non appropriate symbols appear in the units.
Page 2: Line 6 – 4th paragraph (Black paper…): non appropriate symbols appear in the text.
Author Response
Dear Reviewer,
We appreciate your valuable comments. The changed text has been highlighted in green. Please find below the point-by-point responses:
Page 1: Line 4 – 2nd paragraph of the Introduction: non appropriate symbols appear in the units.
RESPONSE: Thank you for the comment. Accordingly, we have made the changes.
¦Ìg/cm2–40.41¦Ìg/cm2 µg/cm2 --> 40.41 µg/cm2
Page 2: Line 6 – 4th paragraph (Black paper…): non appropriate symbols appear in the text.
RESPONSE: Thank you for the comment. Accordingly, we have made the change.
¦Â-caryophylline, --> beta-caryophylline,

Reviewer 2 Report
This manuscript discuss about “Effects of Piper nigrum L. fruit essential oil toxicity against stable fly (Diptera: Muscidae)”. In interesting knowledge has been proposed, however the following comments should be addressed before acceptance
Comments:
1. In abstract section: suggested to add some novelty statement in in abstract section
2. The importance of the study should be clearly mentioned in the introduction section
3. Enhance the quality of images such as Figure1
4. Author should elaborate about the insecticidal activity
5. Author should add brief details about the extraction process of oils form piper nigrum
6. Conclusion should contain more details suggested to add some novel findings
After addressing this comments, manuscript can be acceptable for further progress
Author Response
Dear Reviewer,
We appreciate your valuable comments. The changed text has been highlighted in yellow. Please find below the point-by-point responses:
1. In abstract section: suggested to add some novelty statement in in abstract section
RESPONSE: Thank you for the comment. Accordingly, we have added the novelty statement.
Our findings suggested that, essential oil extracted from P. nigrum fruit could be a potential natural insecticidal agent for stable fly control. To examine the insecticidal properties of P. nigrum fruit EO, further field trials and the efficacy investigation of nano-formulation are required.
2. The importance of the study should be clearly mentioned in the introduction section
RESPONSE: Thank you for the comment. Accordingly, we have added the importance of the study.
The aim of the present study was to evaluate the effects of unripe or green P. nigrum fruit EO insecticidal activities against S. calcitrans using in vitro contact toxicity and fumigant toxicity methods. Thereafter, the EO was analyzed for its chemical composition using gas chromatography-mass spectrometry (GC-MS).
3. Enhance the quality of images such as Figure1
RESPONSE: Thank you for your suggestion. We have made the changes.
4. Author should elaborate about the insecticidal activity
RESPONSE: Thank you for your pertinent comment. We have elaborated the insecticidal activity.
The results showed that S. calcitrans mortality increased with an increase in EO concentration and time.
5. Author should add brief details about the extraction process of oils form piper nigrum
RESPONSE: Thank you very much for the comment. We have added the necessary details.
The EO was extracted from 50 kg of fruit, within 6 h of arrival, by the steam distillation process, where the fruits were immersed in water and the system was heated to 100°C.
6. Conclusion should contain more details suggested to add some novel findings
RESPONSE: Thank you for your suggestion. We have added the novel finding in the Conclusions section.
As there are limited reports available on the insecticidal activities of P. nigrum fruit EO on the horse and livestock vectors such as, horse fly, horn fly, and midges, the study of both P. nigrum fruit EO and nano-formulation for open field trials should be implemented.
